**Cite this article:** Lemanis R. 2020
The ammonite septum is not an adaptation to
deep water: re-evaluating a centuries-old idea.
*Proc. R. Soc. B* **287**: 20201919.

biomechanics, palaeontology, evolution

finite-element analysis, minimum curvature
surface, cephalopods, nanoindentation

**Author for correspondence:**
Robert Lemanis
e-mail: robert_evan.lemanis@tu-dresden.de

Electronic supplementary material is available
online at https://doi.org/10.6084/m9.figshare.
c.5141909.

# The ammonite septum is not an adaptation to deep water: re-evaluating a centuries-old idea

## Robert Lemanis

B CUBE—Center for Molecular Bioengineering, Technische Universität Dresden, Dresden, Germany

RL, 0000-0002-7259-3410

The shells of ammonoid cephalopods are among the most recognizable fossils, whose fractally folded, internal walls (septa) have inspired many hypotheses on their adaptive value. The enduring explanation for their iterative evolution is that they strengthen the shell against pressure at increasing water depths. The fossil record does not definitively support this idea and much of the theoretical mechanical work behind it has suffered from inaccurate testing geometries and conflicting results. By using a different set of mathematical methods compared with previous studies, I generate a system of finite-element models that explore how different parameters affect the shell's response to water pressure. Increasing the number of initial folds of the septa ultimately has little to no effect on the resulting stress in the shell wall or the septum itself. The introduction of higher-order folds does reduce the tensile stress in the shell wall; however, this is coupled with a higher rate of increase of tensile stress in the septum itself. These results reveal that the increase in complexity should not be expected to have a significant effect on the shell's strength and suggests that the evolution of ammonitic septa does not reflect a persistent trend towards deeper-water habitats.

## 1. Introduction

Of the persistent problems in ammonoid palaeobiology, the function of the convoluted ammonite septum is perhaps the most enduring. Since the formal proposal of a structurally supportive function of ammonite septa by Buckland in the 1830s [1], there has been a focus on understanding the shell's capacity to resist hydrostatic pressure while minimizing the amount of material needed to construct the shell. However, over the past 184 years, the actual function of these structures has proven controversial. Indeed, the ammonitic septa exemplifies one of the primary challenges of palaeontology: understanding structures that have no obvious modern analogues.

The archetypical cephalopod shell is divided into two major parts: the body chamber, in which the animal is situated, and the phragmocone (figure 1*a*). The phragmocone, a feature unique to cephalopods [2–4], is formed as the animal grows and deposits a mineralized wall at the rear of the body. Subsequent walls divide the shell tube into a series of discrete chambers that are initially filled with fluid that is gradually replaced with gas, thus creating an internal chamber pressure of around one atmosphere [5–8]. This replacement is performed by the siphuncle (figure 1*a*), a thin organic strand, which is anchored in the first chamber and stretches to the rear of the soft body through all subsequent chambers. This system allows the shell to perform its primary function, buoyancy control, by filling the shell with gas whose volume is unaffected by water pressure. The system, however, results in a significant pressure differential across the wall of the phragmocone.

With this primary function as a backdrop, many authors have attempted to understand the potential adaptive value in the evolution of the progressive frilling

**Figure 1.** Overview of the anatomy and anatomical terms of a computed tomographic dataset of the shell of *Nautilus pompilius* (*a*), a representative cylindrical model created for this study (*b*), and examples of Koch septal surface models (*c*). Septal surfaces are created by using suture line drawings as boundary conditions for the computation of a minimum curvature surface (*c*). The final cylindrical models are formed by combining two cylinders, to form an inner and outer surface, that are capped by hemispherical surfaces to enclose the inner volume. The modelled septal surfaces are then placed inside the cylinders in variable numbers and varying distances to create a range of different morphologies. (Online version in colour.)

of the ammonitic septum. Early septa were dome-shaped [2]; the suture line of these septa (figure 1*b*) would be a simple circle. The beginning of septal folding seems to coincide with the migration of the siphuncle from a more central position to a marginal one [9]. From here, the over 400 Myr of evolution created a broad, iterative trend in Ammonoidea of increasing septal complexity [10], resulting in fractal-like suture lines. There have been numerous potential explanations of evolutionary drivers for this increase in complexity, such as increasing the area of muscle attachments [11–13], increasing the surface area of the supposed gas-secreting tissue [14], improving liquid storage and chamber re-flooding potential [15], and storing small reservoirs of cameral liquid within the small folds of the septa [16–18]. However, the dominant explanation, including the explanation most likely to be encountered in textbooks [19–21], is that it served a strengthening function against external loading [1,22–25].

The full history of functional hypotheses of the ammonitic suture line has been reviewed elsewhere [26,27]. To summarize, this mechanical hypothesis has had several different manifestations, but the three core components are: (1) buttressing the phragmocone wall against indirectly applied loads (i.e. water pressure and predators), (2) supporting the most recently formed septum against direct loads through the body chamber, and (3) increasing the toughness of the structure and allowing the septa to act as springs and decrease bending moments.

Several studies have tested some aspect of these mechanical hypotheses using comparative finite-element analysis (FEA), a computational technique that is capable of modelling how a complex geometry responds under loading [28–30]. This technique has an extensive history in the study of molluscan shell mechanics [31] and cephalopods specifically [25,32–36]. Of particular interest are three papers, two of which produced theoretical models using similar techniques and arrived at conflicting conclusions. Daniel *et al.* [35] argued that increasing septal complexity weakened the shell against external hydrostatic pressure, which was argued against by Hassan *et al.* [25] whose similar models showed the opposite, arguing for the mechanical hypothesis. Another study used empirical models formed from computed tomographic (CT) data to

compare the shells of an ammonite (*Cadoceras* sp.) against *Nautilus* and *Spirula* and found that increasing saddle amplitude weakened rather than strengthened the shell against hydrostatic pressure [36]. This study, however, exemplified a problem in using CT data in that one would need multiple specimens whose morphology was exactly the same with the exception of their septal complexity in order to truly test the effects of different septal morphologies. This is, unfortunately, an impossible task, especially since a number of morphological parameters, including the suture line, of these shells tend to covary with each other [37,38].

In order to maintain strict control over what morphological parameters vary between models, I must return to the mathematical methods discussed previously [25,35]. In doing so, however, the failure of these model to produce consistent results has to be addressed. In the work of both Daniel *et al.* [35] and Hassan *et al.* [25], septa were generated mathematically via a summation of multiple 2D Fourier series to generate a 3D surface. In this case, the end result morphology has no morphological 'control' and some generated morphologies are not similar to true septa [25].

Some differences between the models include discretization, which has been considered a source of, at least, some of the error [25]. Daniel *et al.* [35] created models composed of four-noded, flat plate elements. Hassan *et al.* [25] composed their models from eight-noded curved shell elements. The curvature of the septa, in terms of the lobes and saddles, also extended to the centre of the septum in the models of Daniel *et al.* [35] while the centres were smoothed in the models of Hassan *et al.* [25], which is the more accurate approximation of true septal morphology.

To overcome the errors in the previous modelling methods, I developed a different methodology to construct the shell and septa. The method employed here uses an initial suture line to create the septa as a minimum curvature surface, first proposed by Hammer [39], that recreates any arbitrary septal frilling and maintains a realistic surface curvature. A series of cylindrical theoretical models (figure 1*b*) with varying morphological parameters are developed to test their effects on the shell's response to hydrostatic pressure. The system of morphological parameters: shell

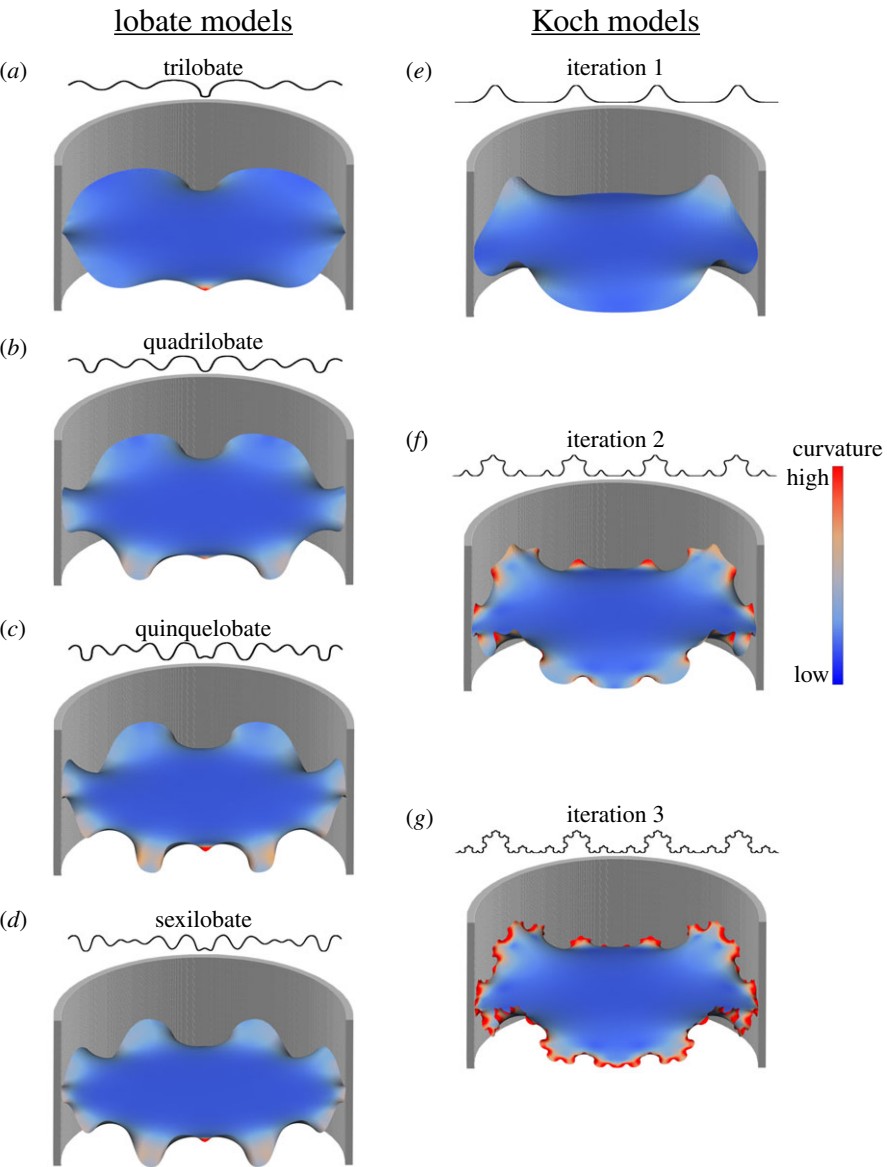

**Figure 2.** Lobate (*a–d*) and Koch (*e–g*) septal models. These models were created by wrapping the shown respective curves (simulated suture lines) around a cylinder and using these closed curves as boundary conditions for the approximation of a minimum curvature surface. Lobate curves are based off of the four primary suture lines presented by Korn *et al.* [40]. Koch models are constructed using generated Koch curves, of 1–3 iterations. The Koch curves are duplicated four times to create four total lobes. Mapped onto each septal surface is a curvature map showing the overall low curvature of the septal surface and the rapid decrease of curvature towards the centre even at higher iterations. (Online version in colour.)

thickness, septal spacing and sutural complexity (figure 2) are all tested against simulated water pressure.

## 2. Material and methods

### (a) Indentation

Nanoindentation of a cross section of the shell of *Nautilus pompilius* was performed with a Triboscan TI 950 with a Berkovich diamond tip. A maximum load of 5000 µN was applied with a load function consisting of a 10 s loading segment, a 10 s holding segment and a 10 s unloading segment. Reduced modulus and hardness data was collected from a 320 point grid with a 5 µm spacing between points (electronic supplementary material, table S1). The polished sections consisted of the outer spherulitic-prismatic layer and the nacre layer but excluded the inner prismatic layer.

### (b) Modelling

All base models, including the cylinders and septa, were constructed in Rhino (v5/6, McNeel; https://www.rhino3d.com/).

Simple hemispherical caps were created for each cylinder to completely enclose the interior (figure 1*b*) that both satisfies the assumptions for the theoretical pressure vessel calculations as well as simplifies the FEA boundary conditions by removing the need to constrain the open edges of the cylinders.

Half suture lines were mirrored to form a complete, bilaterally symmetric suture line. Smooth Koch curves are generated using splines of up to three iterations (http://fractalcurves.com/app/) and duplicated to create four lobes (figure 2*e–g*). In both cases, once the complete suture line is assembled an image is saved and imported into Inkscape (v. 0.92, https://inkscape.org/) where they are traced over and exported as vector image files. These files are imported into Rhino and wrapped around a cylinder and the two ends of the curve are joined together to form a single, closed curve. Septal surfaces (figure 2) were constructed using an algorithm, implemented in grasshopper, which approximates a minimum curvature surface based on the single, closed curve (https://github.com/Mathias-Fuchs).

Two septal surface meshes were lofted together and joined after their conversion to NURBs surfaces. These joined structures were then combined with the inner cylinder via a Boolean union,

all of which was performed with Rhino. The entire structure was meshed and exported to Avizo (Material Science, v. 9.7, http://www.vsg36.com/) where the mesh was refined, remeshed and remaining defects were manually corrected. These meshes were then imported into Gmsh [41] where they are meshed into tetrahedral meshes with quadratic elements using the Frontal algorithm. Tetrahedral meshes are then imported into Abaqus (2016, Dassault Systèmes Simulia; https://www.3ds.com/products-services/simulia/products/abaqus).

### (c) Finite-element analysis

All models were treated as linear, elastic and loaded with a 2 MPa pressure load over the entire external surface. The models have an isotropic elastic modulus of 70 GPa, based on nanoindentation of the shell of *Nautilus pompilius*, and a Poisson's ratio of 0.3, a typically accepted value for molluscan shells [31]. The Abaqus output was transformed into cylindrical coordinates in order to calculate the cylindrical stress components: radial (parallel to the radius of the cylinder), tangential (hoop, perpendicular to the radius) and longitudinal (parallel to the long axis of the cylinder) stress. Validation and error calculations were done on the basis of comparison with theoretical hoop stress values for a thin walled pressure vessel

$$\sigma_h = \frac{Pr}{t},$$

where $\sigma_h$ is the hoop stress, $P$ is the pressure, $r$ and $t$ are the radius and thickness of the cylinder, respectively. Theoretical hoop stress for the 0.5 mm cylinder is 36.4 MPa and 182.2 MPa for the 0.1 mm cylinder. However, the models are not hollow cylinders due to the presence of the septa, and therefore do not fully comply with the assumptions of the equation. Proper areas to take values from needed to be identified and were ultimately taken from areas on the cylinder farthest from the septa and prior to the hemispherical caps. For all models, calculated error ranged from 0.1% to 4%.

## 3. Results

### (a) Nanoindentation

Indentation hardness and reduced modulus values (electronic supplementary material, table S1) closely follow the ultrastructural features of *N. pompilius*. The highest reduced modulus values occur in the spherulitic-prismatic layer, ranging from 60 to 95 GPa. The transition zone between the spherulitic-prismatic layer and nacre shows the lowest modulus values, from 50 to 60 GPa. This zone also shows an apparently greater porosity as well as irregularly shaped nacre tablets [42]. Reduced modulus increases again to 60–80 GPa in the nacre layer.

### (b) Modelling

A total of 11 models were constructed to test the effects of shell thickness, septal complexity and septal spacing. The developed general method to construct all models was the same with slight alterations depending on the required thicknesses and the type of septa required, divided between 'lobate' and 'Koch' septal models. Lobate models are septal models whose governing suture line is derived from one of the four primary suture lines [40]: trilobate, quadrilobate, quinquelobate and sexilobate (figure 2a–d). Koch model suture lines are derived from standard fractal Koch curves of 1–3 iterations. The generated Koch curves were duplicated to create the equivalent of a four lobed suture line (figure 2e–g). For both sets of models, the final suture line was then wrapped around a cylinder, with a radius of 9.11 mm, to create a circular suture line. This closed line was then set as a boundary condition for the calculation of a minimum curvature surface.

The shell wall was simulated using two cylinders to form the inner and outer walls. The inner cylinder had a radius of 9.11 mm, and the outer cylinder had a radius of either 9.21 mm, for the 0.1 mm shell thickness models, or 9.61 mm, for the 0.5 mm shell thickness models. The lobate models have a septal thickness of 0.4 mm and the Koch models have a 0.04 mm thickness due to the fact that increasing the thickness of the Koch models causes the suture lines to self-intersect. After meshing, all models were loaded with a 2 MPa pressure over their external surface (electronic supplementary material, tables S2 and S3). Several additional analyses were performed with the Koch models at 1 and 3 MPa pressures (electronic supplementary material, table S3).

### (c) Septal spacing

The effects of septal spacing was tested by comparing three models with the same sexilobate septal morphology (figure 3). Septal distances of 5 (5 septa), 10 (3 septa) and 20 (2 septa) millimetres were compared. For quantitative comparisons, I compare extracted maximum principal stress values and tangential stress values from a line probe, parallel to the longitudinal axis of the cylinder, which runs along the outer surface of the cylinder (figure 1b).

Septal spacing has little to no effect on overall stress magnitudes (figure 3a–c). The septa do have a noticeable local effect on stress distribution, where the more tightly packed septa seem to prevent the formation of a relatively higher, circumferential stress region forming between the septa seen in the model with a septal distance of 10 mm (the darker blue band between septa; figure 3b). The larger region between the two septa in the 20 mm model also lacks this higher stress region on the exterior of the shell, instead small areas of roughly equal stress form in the 'shadow' of the septal concavities with this higher stress band forming on the interior of the shell (figure 3c). Septal spacing therefore changes the stress by shifting the bending pattern of the shell around the septa under hydrostatic pressure. Comparing the nodal displacements, the more tightly packed, 5 mm spaced, septa minimize the maximum displacement of the shell wall between the septa while in both the 10 and 20 mm spacing models the displacements of the shell wall are much the same as the displacements of the unsupported shell wall (figure 3d–f). This suggests there exists some minimum interseptal distance where this displacement minimization effect can occur. This effect is also seen by comparing the total strain energy of the models, which decreases as septal spacing increases (electronic supplementary material, table S4).

### (d) Shell thickness

Trilobate and sexilobate models were tested with shell thicknesses of 0.1 mm and 0.5 mm (figure 4). The 0.1 mm thick shell shows drastically higher stress values compared with the 0.5 mm thick shell. An 80% decrease in shell thickness here translates to a 433–467% increase in tangential stress. A similar increase in maximum principal stress is also observed, increasing by 461–603%. It should be noted that the theoretical hoop stress calculated using the equation

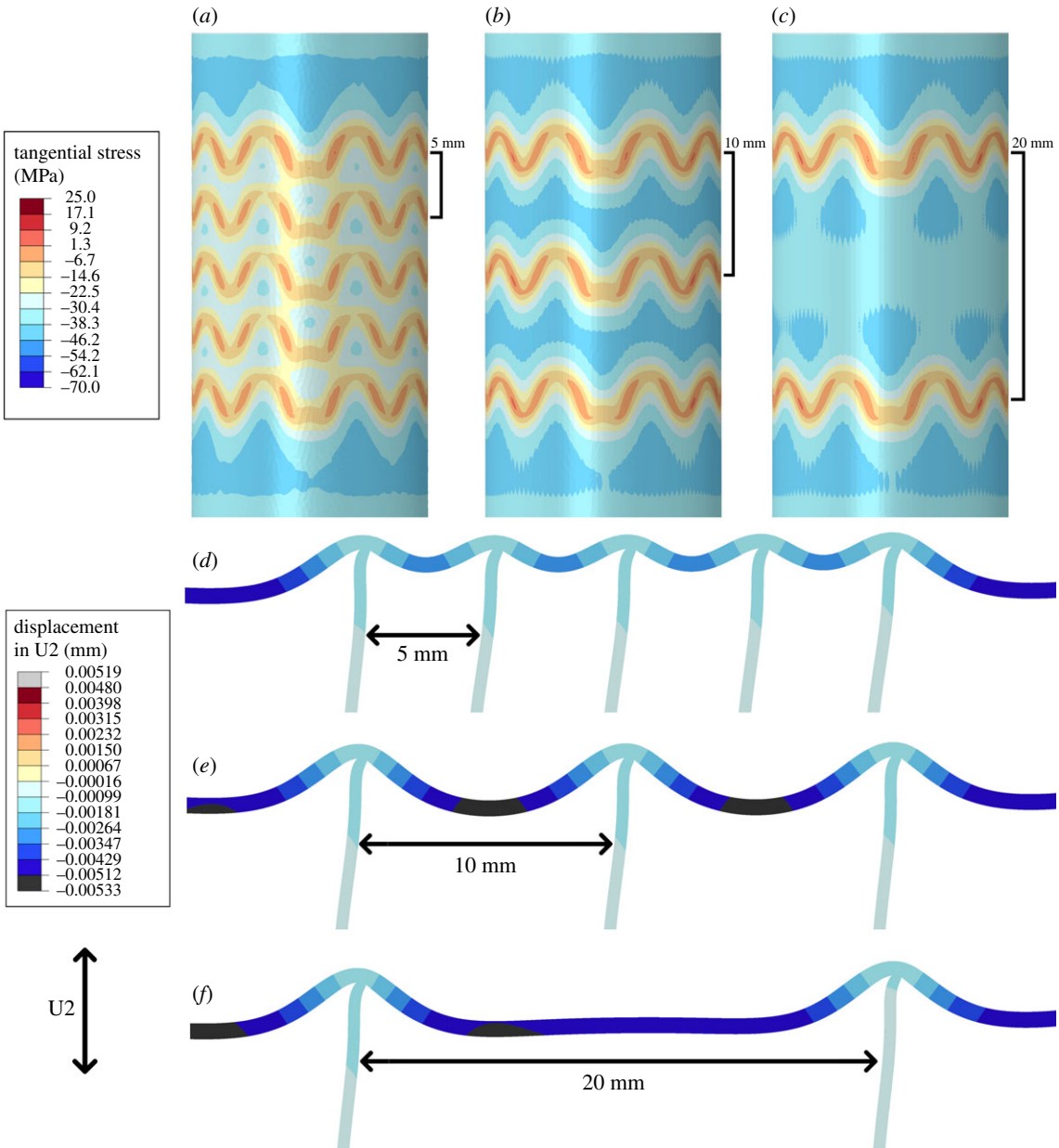

**Figure 3.** Comparisons of the tangential (hoop) stress contours (*a–c*) and deformation (*d–f*) between three models with different septal spacing values: 5 mm (*a, d*), 10 mm (*b,e*) and 20 mm (*c,f*). The single plane slices show a greatly exaggerated deformation (in millimetres) under hydrostatic pressure. Changing the values of septal spacing do not noticeably alter the magnitude of tangential stress (or maximum principal stress) between the models. However, the more closely packed septa limit the total displacement of the surrounding shell wall (*d–f*) and the values of stress between septa (*a*). This effect disappears in the 10 mm spacing model suggesting that the critical value for this effect is between 5 and 10 mm, which also correlates to the disappearance of the darker (blue) band of higher stress (seen in (*b*)) in the 5 mm spacing model. (Online version in colour.)

above for a hollow cylinder, increases from 36.44 MPa to 182.2 MPa—an increase of 400%.

The overall pattern of stress formation does not significantly change with shell thickness. On the exterior surface, the highest magnitude tangential stresses occur along the shell wall, between septa and within the concave folds of the suture line (figure 4). On the interior surfaces, high magnitude stress forms along the shell wall, and along the flanks of the larger lobes and saddles. Maximum principal stress, similar to empirical cephalopod models [36], shows peaks along the septa-shell wall attachment zone.

## (e) Lobate models

The main effect of increasing the complexity of the primary suture line is the redistribution of the stress contours in the shell and septa with minimal change in their magnitude (figure 5*a–h*). The complexity of the primary suture line is simply defined here as the number of lobes, beginning with the trilobate suture line as the most simple and the sexilobate suture line as the most complex. Comparing tangential stress and maximum principal stress taken from the line probe, there is no regular pattern in terms of increasing/decreasing stress values against septal complexity (figure 5*i*). The variation of stress values around equivalent septa on the different models is due to the redistribution of stress, which can be seen in the contour maps (figure 5*a–d*).

Comparing the stress values around the middle septum, the overall ranges in both tangential and maximum principal stress are similar in all models. Compressive stress is highest along the shell wall with less negative values and tension forming along the suture line. Around the middle septum,

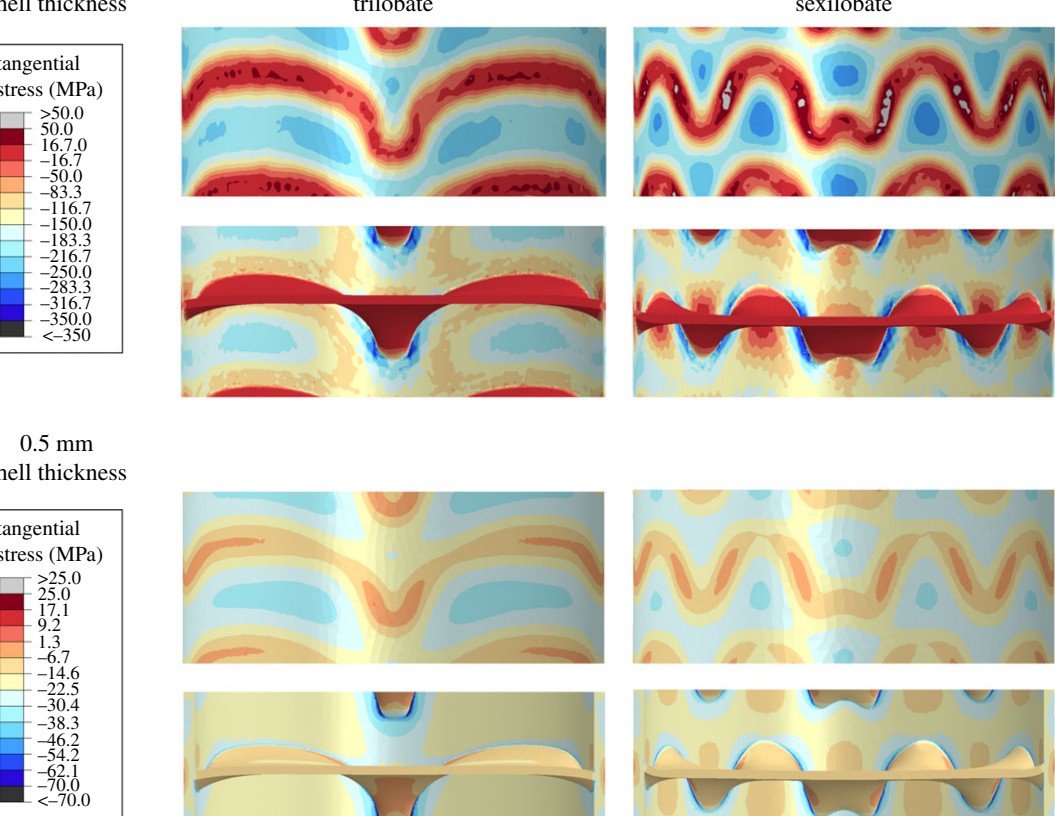

**Figure 4.** Comparison of tangential stress contours between the 0.5 mm and 0.1 mm shell thickness models. Tangential stress values decrease towards the suture line, going from compressive (negative) to tensile (positive) and reach a maximum magnitude between septa (within the darker blue contours). This comparison demonstrates that shell thickness is the most significant variable determining stress magnitude on the shell with septal complexity (illustrated via trilobate and sexilobate models) having little effect on stress magnitude compared with the shell wall thickness. The strong effect of shell thickness challenges the idea that increasing septal complexity can compensate for decreasing shell thickness, which is illustrated in the comparison between the thinner, sexilobate model against the thicker, trilobate model. (Online version in colour.)

the quadrilobate model shows the most negative tangential stress at the point of the suture (figure 5f); the sexilobate model shows slightly less negative stress compared with the quinquelobate and trilobate models (figure 5i). Similarly, at the suture line, the sexilobate model shows the highest value of maximum principal stress (9.1 MPa) while the quadrilobate models shows the lowest stress (0.92 MPa).

## (f) Koch models

Progressively increasing the order of folds (figure 1c) redistributes stress in the area sound the septum and the suture line with some change in total stress magnitude and total strain energy (figure 6; electronic supplementary material, table S4). The third iteration model has the highest magnitude tangential stress from the line probe, although this value is only 3.85 MPa higher than the peak magnitude value from the second iteration model. Contour maps show that the peak stresses tend to develop in areas of highest curvature (figures 2e,f and 6). The second and third iteration models have tighter curves along the flanks of the lobes that concentrate stress at a higher level than the first iteration model (figure 6d–f). There is a small effect seen when comparing the second and third iteration models where the peak stress zone is reduced in the third iteration model compared with the second (figure 6b,c). Besides the line probe, additional

points that lie within the maximum contours on the shell wall and on the septum can also be compared. The average maximum principal stress on the external shell wall decreases with increasing complexity by 7.8 MPa between the first and third iteration models with a decrease in peak maximum principal stress of 12.3 MPa. However, comparing values from the septa themselves, there is an average increase in maximum principal stress going from the first to third iteration models. The average maximum principal stress increases by 19.12 MPa and the peak maximum principal stress increases by 22.2 MPa (table 1).

A consistent effect of increasing the complexity of both the Lobate and Koch models is a general reduction of the displacement of the septa, specifically in the centre of the septum. The displacement at the centre of the first iteration model is around 77 µm while in the third iteration model the displacement is around 30 µm. Similarly, in the trilobate model the displacement of the centre of the septum is between 6.5 and 9 µm compared with the sexilobate model that is between 3.7 and 7 µm. The difference between the lobate and Koch model displacements can be attributed to the differences in septal thickness. Displacements of the shell wall do not show the same significant change between septal morphologies. Also noted are small decreases in total model strain energy as septal complexity increases (electronic supplementary material, table S4).

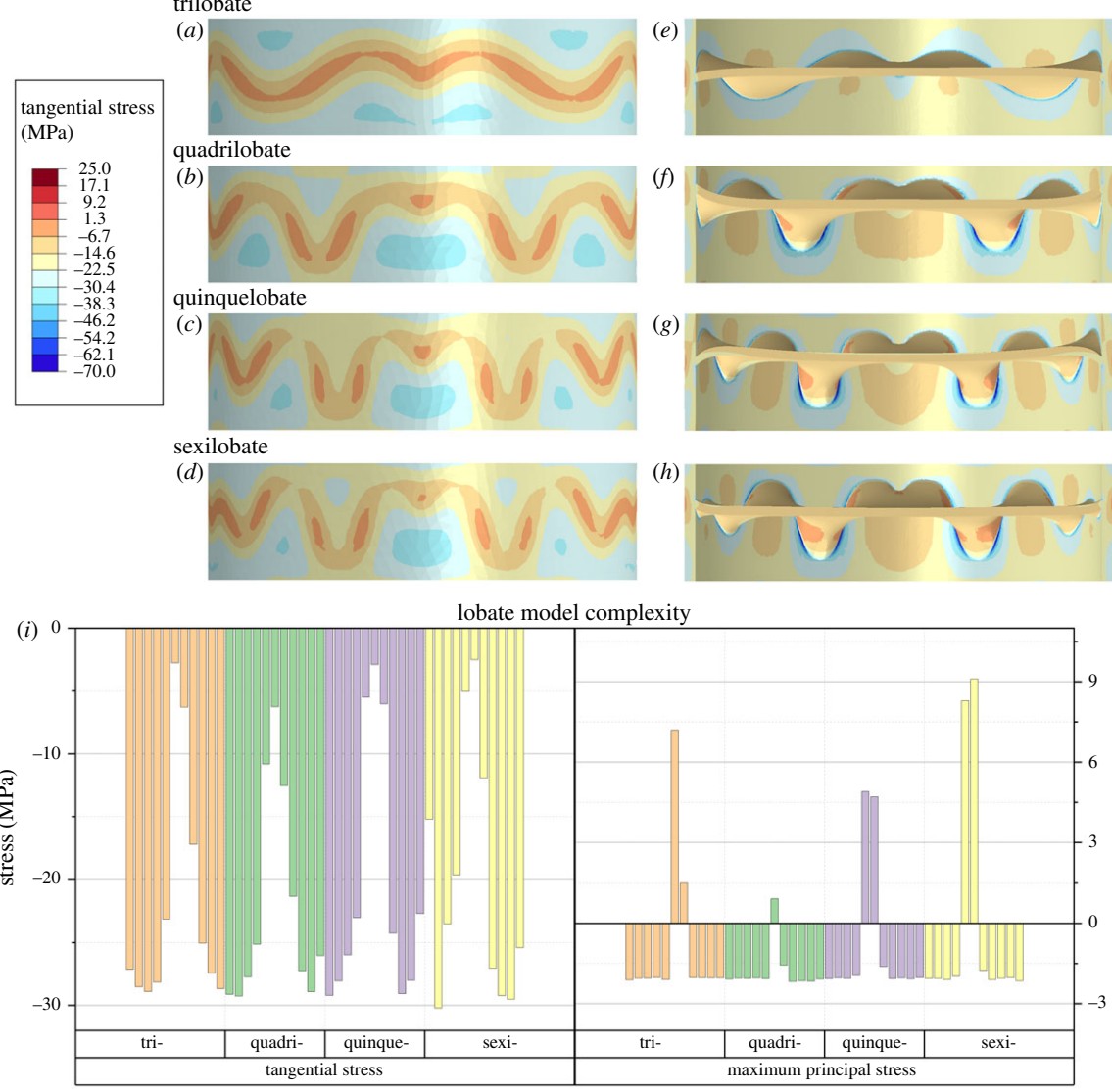

**Figure 5.** Tangential stress contours about the middle septum in the four lobate models: trilobate (*a,e*), quadrilobate (*b,f*), quinquelobate (*c,g*) and sexilobate (*d,h*). Contours on both the external shell face (*a–d*) and the internal shell and septa (*e–h*) are shown. The highest magnitude stresses occur along the inner wall, along the flanks of the flutes where the septa connect to the shell. Tangential stress and maximum principal stress values are also extracted from a line probe along the length of the model along the exterior shell wall (figure 1*b*) for all lobate models (*i*). The presented values are taken from the area around the middle septum (of the five total septa). Less negative tangential stress values and higher maximum principal stress values occur at the suture line. (Online version in colour.)

## 4. Discussion

Our results directly challenge the common explanation for a long-standing palaeontological mystery. The developed models allow us a degree of control over the final expressed morphology that computed tomographic data of real shells would not allow. While similar models have been produced in the past [25,33,35], the directly contradictory results from some of these analyses indicate either modelling or analytical errors that required a reevaluation of the techniques used in their construction.

To add some context to our results, an evaluation about how the models presented here are different than previous work is necessary. The two previous studies create models with only two septa within the shell tube [25,35], which the results here have shown could be problematic. The local loading configuration of septa is naturally non-symmetric due to differences between the two flute types (lobes versus saddles; figure 1*c*). This means that the two exterior most septa will have different resulting values (stress, strain and displacement); furthermore, they will also show different values compared with the septa

between them as they change the local loading conditions of these inner septa. The result of this system is that the two exterior septa are not adequate models of an arbitrary septum within the phragmocone that would only be loaded via lateral pressure from the shell wall, which is why results presented here are taken from the middle septum (figures 4–6).

The resulting stress patterns between all three studies are notably different. Peak maximum principal stresses in our models occur along the suture line in all complexities and along the flanks of the flutes, similar to the results from empirical models [36], rather than just the flanks [25] or along the flute fold axis [35]. Interestingly, both prior studies note the concentration of high stress towards the centre of the septum with increasing complexity [25,35], which is not seen in any of the models shown here. Comparing the lobate models, there is an increase in the average tangential stress in the sexilobate model compared with the other models though the quinquelobate model rather than the trilobate model shows an overall lower average stress magnitude across the septal face (electronic supplementary material, figure S1).

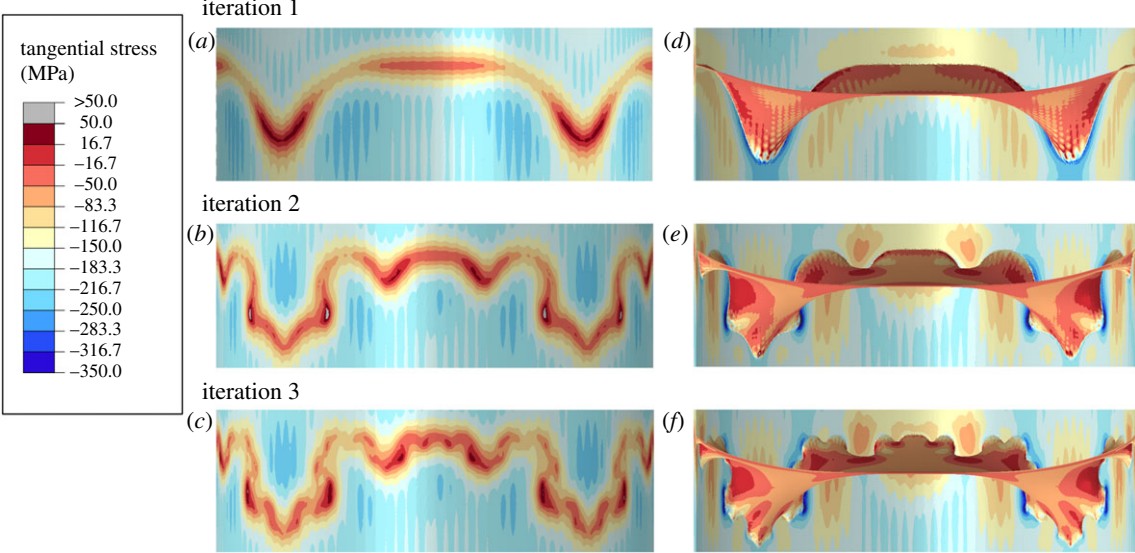

**Figure 6.** External (*a–c*) and internal (*d–f*) comparisons of stress contours between the three Koch models with an increasing order of complexity: starting with first-order folds (*a,d*) and increasing to second order (*b,e*) and third order (*c,f*). Increasing the order of folding has limited changes on changing the stress magnitudes. Local increases occur along the suture line where they occur at the tips of the second- and third-order folds (*b,c,e,f*). (Online version in colour.)

**Table 1.** Average maximum principal stresses (±1 s.d.).

|  | iteration 1 | iteration 2 | iteration 3 |
|---|---|---|---|
| shell wall |  |  |  |
| 1 MPa | 21.91 (± 15.7) | 18.73 (±14.1) | 17.75 (±12.7) |
| 2 MPa | 44.75 (±32.2) | 39.88 (±29.7) | 36.95 (±25.6) |
| 3 MPa | 66.18 (±47.3) | 58.47 (±43.3) | 55.70 (±38.3) |
| septum |  |  |  |
| 1 MPa | 6.32 (±7.33) | 11.03 (±9.38) | 17.10 (±9.81) |
| 2 MPa | 10.33 (±13.2) | 20.34 (±19.1) | 29.45 (±18.4) |
| 3 MPa | 15.88 (20.73) | 32.40 (±19.10) | 47.40 (±27.82) |

The results presented here show that, at best, increasing complexity of the septa does have a slight positive effect on the shell wall's strength against hydrostatic pressure, though this is coupled with a weakening of the septum itself (table 1). However, at worst, some complex forms, such as the second iteration model, show higher tangential stress values along the shell wall and suture that would potentially fail before the first iteration model. Also notable is that with increasing pressure, the rate of increase of tensile stress in the more complex septa is higher than the rate of decrease of stress in the shell wall (table 1).

Shell wall thickness, over any of the other morphological parameters, is the primary determinant in the stress magnitudes in the models (figure 4). It seems very unlikely that increasing septal complexity could compensate for a general thinning of the shell, as has been previously proposed [25,43]. If this relationship was true, it would also mean the body chamber would be thinner as well, and therefore weaker, in forms with more complex septa. Therefore, one might expect a greater proportion of reported shell injuries in forms with complex septa. This follows from the observation that shell repair of the phragmocone is generally not possible [44], and remodelling and resorption are uncommon in cephalopods outside of resorption of some external

ornamentation that occurs in the whorl overlap region [45]. This is unsurprising since the septa limit the rear movement of the soft body and prevent large-scale remodelling of the previous whorls, as seen in some gastropods [46], as well as any potential repairing of the phragmocone. However, there is no positive correlation between the proportion of shell injuries and septal complexity [47].

Similarly, increasing septal complexity does seem to decrease the overall deflection of the septa. This decrease of deflection with increasing complexity is not seen in the shell wall; the shell still bends around the septa and generates the highest bending moments along the suture line regardless of the order of folds present (figure 6). Though the magnitude of this bending of the shell wall is slightly lower with higher-order folds (table 1). While I do not calculate the bending moments directly, tensile and compressive stress magnitudes should correlate proportionally with bending moments.

Along with decreasing displacement, there is also a trend of decreasing maximum principal stress going from the first iteration Koch model to the third iteration model (table 1). While this effect is small, and the total range of maximum principal stress values in these regions tends to overlap between all models, it is interesting to calculate how this difference translates into depth. By calculating simple linear trend lines for both the first and third iteration models, I can estimate how much additional pressure would be needed to elevate the stress in the shell wall of the second and third iteration model to the same magnitude as the first iteration model. With the first iteration model at 2 MPa, the second iteration model would need to be under a pressure of 2.25 MPa and the third iteration model would be under 2.38 MPa, an increase of around 25 m and 38 m depth, respectively. This small increase in depth tolerance, viewed in reference to the shell wall, is somewhat counteracted by the increase in maximum principal stress in the septa, especially at deeper depths since the rate of increase of this stress in the septa is higher than the rate of decrease in the shell wall. This trend is also not the same for tangential stress as this stress increases in the second and third iteration models at the exterior surface of the shell wall (figure 6*a–c*).

These results raise the question as to what association is seen in the fossil record between septal complexity and facies. The relationship between these parameters is complex, and there is no simple, universal correlation between complexity and depth [10]. Multiple analyses of septal complexity in Jurassic ammonites has shown no significant correlation between complexity and general environment, which was divided into neritic (more coastal, shallow water) and epioceanic (open waters away from the continental shelf), arguing for either a lack of correlation between habitat depth and complexity or a consistent habitat depth in both environments [48,49]. Indeed, in a very general sense, it seems ammonoids inhabited the upper 250 m of the water column [10] and it seems unlikely that the evolution of the ammonitic septum was driven by migration into deeper waters.

Data accessibility. Nanoindentation and finite-element analysis data are included as electronic supplementary material, information. The code for the algorithm to generate minimum curvature surfaces is freely available from Github: https://github.com/Mathias-Fuchs/MinimalSurface

Competing interests. I declare I have no competing interests.

Funding. This work was supported by the Deutsche Forschungsgemeinschaft through grant LE 4039/1-1.

Acknowledgements. Mathias Fuchs is thanked for his help working with the algorithm and bug fixes. Igor Zlotnikov provided helpful suggestions and improved this paper. I thank C. Klug and one anonymous reviewer for their helpful reviews.

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
