## [Reviewer comments · Proceedings of the Royal Society B: Biological Sciences]

Review History

RSPB-2020-1919.R0 (Original submission)

Review form: Reviewer 1 (Christian Klug)

Recommendation

Accept with minor revision (please list in comments)

Scientific importance: Is the manuscript an original and important contribution to its field?

Excellent

General interest: Is the paper of sufficient general interest?

Good

Quality of the paper: Is the overall quality of the paper suitable?

Acceptable

Is the length of the paper justified?

Yes

Should the paper be seen by a specialist statistical reviewer?

No

Do you have any concerns about statistical analyses in this paper? If so, please specify them explicitly in your report.

No

It is a condition of publication that authors make their supporting data, code and materials available - either as supplementary material or hosted in an external repository. Please rate, if applicable, the supporting data on the following criteria.

Is it accessible?

Yes

Is it clear?

Yes

Is it adequate?

Yes

Do you have any ethical concerns with this paper?

No

Comments to the Author

Dear Robert, dear editors,

thank you for considering me as reviewer of this interesting manuscript.

As always, Robert, you apply sophisticated methods and provide interesting results. In congratulate for this.

Also, I fully agree with both the research question and your conclusion.

Nevertheless, there are some smaller issues. I have marked them in the pdf and repeat some here:

1. The abstract needs attention. It is sometimes not entirely clear what kind of stress you talk about.

2. Page 4: You list 3 core components: this list lacks your suggestion from 2016 that the frilling may increase resistance against point loads by predators' teeth and jaws.

It also lacks the null-hypothesis that the frilling is simply fabrication noise.

3. Why do you use the 9.11 mm for the cylinder?

4. You repeatedly refer to the papers by Daniel and Hassan (you do cite it), but you do not provide the reference at all the places. It is clear to me but maybe not to all readers.

5. You repeatedly write "we", but you are the single author, so replace by "I" throughout.

6. Last phrase of conclusion: While I agree with your conclusion, it would be helpful to know your opinion why ammonite sutures ARE frilled. My thought:

I think that we still have 2 reasonable interpretations:

A. The reduction of ammonitella-size implies a greater degree of size-change in the spetal surface. With each new septum, it has the chance to increase complexity. we see this link in ammonoids and nautilids!

B. Potentially, it enables higher growth rates because the new septa that are not yet fully mineralized might be more tightly stayed when they are more strongly frilled than simpler septa. This is something that needs to be tested, maybe your next paper?

For further comments, see the pdf!

Besides, interesting work, keep it up!

Best wishes,

Christian

Review form: Reviewer 2

Recommendation

Accept with minor revision (please list in comments)

Scientific importance: Is the manuscript an original and important contribution to its field?
Excellent

General interest: Is the paper of sufficient general interest?
Good

Quality of the paper: Is the overall quality of the paper suitable?
Excellent

Is the length of the paper justified?
Yes

Should the paper be seen by a specialist statistical reviewer?
No

Do you have any concerns about statistical analyses in this paper? If so, please specify them explicitly in your report.
No

It is a condition of publication that authors make their supporting data, code and materials available - either as supplementary material or hosted in an external repository. Please rate, if applicable, the supporting data on the following criteria.

Is it accessible?
N/A

Is it clear?
Yes

Is it adequate?
N/A

Do you have any ethical concerns with this paper?
No

Comments to the Author

This is a thorough and compelling manuscript, and will be a valuable contribution to the history of this discourse. The new observations are much needed, and the interpretations are appropriate and within the scope of the stated objectives.

I suggest minor revisions that might aid reader appreciation and interfacing with related biological interests.

The first is a matter of discussion that might be boosted by adding a paragraph or expanding what is already here. Biomineralization in molluscs is intensely studied (by biologists) and complex in execution (by mollusks themselves!). I understand from the text that the author has strong reasoning to convince the reader that shell thickness is not really the deciding factor in these matters; two shell thicknesses are presented and the fundamental results are discussed in other directions. But there's a matter buried in here that piques my curiosity. The author posits that a thin shell along the flanks of the phragmacone would require a matching thin shell along the flanks of the body chamber. From first principals of shell formation and growth this makes sense, since the venter and flanks of today's body chamber become the phragmacone of tomorrow's larger adult. But... did ammonites rework, re-dissolve, re-mineralize, etc the aragonite within their shell interior surfaces? Plenty of molluscs do! I do not bring this up as a bugaboo - it's a sincere question to which I don't know the answer, and it seems central to the main conceit by which shell thickness is dismissed in the Discussion. So the simple revision I request is this - can the author do a bit of leg work here for us? Are there observations that strongly support or reject

ammonite reworking of internal aragonite surfaces on par with other molluscs? Or is this an area of insufficient knowledge? Either way, I think mention of this issue belongs in the intro/methods (when we are informed of the target shell thicknesses) and in the Discussion (when the author weighs the pros/cons of thickness for the function/growth/development of the animal). Again, I do not think the answer to this query will hinder the overall paper - I just think it's due diligence to include in Discussion.

Secondly, I recommend revision of adding one or more figures to the paper. I know the historic and paleontological significance of this work and it is a valued contribution, but I want a non-specialist to appreciate what's at stake here. I think one or two small figures would help. Something on the order of a cartoon that presents the premise at the beginning, and the conclusion at the end. Think of the figures that pepper papers from the Cretaceous work in the 1980s that depict the depth distributions of ammonites; these grab reader's attention and end up in text books. It's almost a "look what we can do! stick them in water profiles!" Ok, well, the author's discussion here handles many references to associations (or lack there of) between water/habitat depth, time, and suture complexity. Can we get a cartoon simplified illustration of this? Perhaps as a timeline, or as a water column profile? A little diagram like that might go a long way in the mind of a reader, in terms of setting up the problem and providing your answer to it.

Overall the work is valuable and interesting, and I encourage the Editor to welcome minor revisions.

Decision letter (RSPB-2020-1919.R0)

14-Sep-2020

Dear Dr Lemanis

I am pleased to inform you that your Review manuscript RSPB-2020-1919 entitled "The ammonite septum is not an adaptation to deep water: re-evaluating a centuries old idea" has been accepted for publication in Proceedings B.

The referee(s) do not recommend any further changes. Therefore, please proof-read your manuscript carefully and upload your final files for publication. Because the schedule for publication is very tight, it is a condition of publication that you submit the revised version of your manuscript within 7 days. If you do not think you will be able to meet this date please let me know immediately.

To upload your manuscript, log into <http://mc.manuscriptcentral.com/prsb> and enter your Author Centre, where you will find your manuscript title listed under "Manuscripts with Decisions." Under "Actions," click on "Create a Revision." Your manuscript number has been appended to denote a revision.

You will be unable to make your revisions on the originally submitted version of the manuscript. Instead, upload a new version through your Author Centre.

- 1) A text file of the manuscript (doc, txt, rtf or tex), including the references, tables (including captions) and figure captions. Please remove any tracked changes from the text before submission. PDF files are not an accepted format for the "Main Document".

2) A separate electronic file of each figure (tiff, EPS or print-quality PDF preferred). The format should be produced directly from original creation package, or original software format. Please note that PowerPoint files are not accepted.

3) Electronic supplementary material: this should be contained in a separate file from the main text and the file name should contain the author's name and journal name, e.g. `authorname_procb_ESM_figures.pdf`

All supplementary materials accompanying an accepted article will be treated as in their final form. They will be published alongside the paper on the journal website and posted on the online figshare repository. Files on figshare will be made available approximately one week before the accompanying article so that the supplementary material can be attributed a unique DOI. Please see: <https://royalsociety.org/journals/authors/author-guidelines/>

4) Data-Sharing and data citation

It is a condition of publication that data supporting your paper are made available. Data should be made available either in the electronic supplementary material or through an appropriate repository. Details of how to access data should be included in your paper. Please see <https://royalsociety.org/journals/ethics-policies/data-sharing-mining/> for more details.

<http://datadryad.org/submit?journalID=RSPB&manu=RSPB-2020-1919> which will take you to your unique entry in the Dryad repository.

Once again, thank you for submitting your manuscript to Proceedings B and I look forward to receiving your final version. If you have any questions at all, please do not hesitate to get in touch.

Sincerely,

Dr Daniel Costa

Reviewer(s)' Comments to Author:

Referee: 1

Comments to the Author(s)

Dear Robert, dear editors,

thank you for considering me as reviewer of this interesting manuscript.

As always, Robert, you apply sophisticated methods and provide interesting results. In congratulate for this.

Also, I fully agree with both the research question and your conclusion.

Nevertheless, there are some smaller issues. I have marked them in the pdf and repeat some here:

1. The abstract needs attention. It is sometimes not entirely clear what kind of stress you talk about.

2. Page 4: You list 3 core components: this list lacks your suggestion from 2016 that the frilling may increase resistance against point loads by predators' teeth and jaws.

It also lacks the null-hypothesis that the frilling is simply fabrication noise.

3. Why do you use the 9.11 mm for the cylinder?

4. You repeatedly refer to the papers by Daniel and Hassan (you do cite it), but you do not provide the reference at all the places. It is clear to me but maybe not to all readers.
5. You repeatedly write "we", but you are the single author, so replace by "I" throughout.
6. Last phrase of conclusion: While I agree with your conclusion, it would be helpful to know your opinion why ammonite sutures ARE frilled. My thought:

I think that we still have 2 reasonable interpretations:

A. The reduction of ammonitella-size implies a greater degree of size-change in the spetal surface. With each new septum, it has the chance to increase complexity. we see this link in ammonoids and nautilids!

B. Potentially, it enables higher growth rates because the new septa that are not yet fully mineralized might be more tightly stayed when they are more strongly frilled than simpler septa. This is something that needs to be tested, maybe your next paper?

For further comments, see the pdf!

Besides, interesting work, keep it up!

Best wishes,

Christian

Referee: 2

Comments to the Author(s)

This is a thorough and compelling manuscript, and will be a valuable contribution to the history of this discourse. The new observations are much needed, and the interpretations are appropriate and within the scope of the stated objectives.

I suggest minor revisions that might aid reader appreciation and interfacing with related biological interests.

The first is a matter of discussion that might be boosted by adding a paragraph or expanding what is already here. Biomineralization in molluscs is intensely studied (by biologists) and complex in execution (by mollusks themselves!). I understand from the text that the author has strong reasoning to convince the reader that shell thickness is not really the deciding factor in these matters; two shell thicknesses are presented and the fundamental results are discussed in other directions. But there's a matter buried in here that piques my curiosity. The author posits that a thin shell along the flanks of the phragmacone would require a matching thin shell along the flanks of the body chamber. From first principals of shell formation and growth this makes sense, since the venter and flanks of today's body chamber become the phragmacone of tomorrow's larger adult. But... did ammonites rework, re-dissolve, re-mineralize, etc the aragonite within their shell interior surfaces? Plenty of molluscs do! I do not bring this up as a bugaboo - it's a sincere question to which I don't know the answer, and it seems central to the main conceit by which shell thickness is dismissed in the Discussion. So the simple revision I request is this - can the author do a bit of leg work here for us? Are there observations that strongly support or reject ammonite reworking of internal aragonite surfaces on par with other molluscs? Or is this an area of insufficient knowledge? Either way, I think mention of this issue belongs in the intro/methods (when we are informed of the target shell thicknesses) and in the Discussion (when the author weighs the pros/cons of thickness for the function/growth/development of the animal). Again, I do not think the answer to this query will hinder the overall paper - I just think it's due diligence to include in Discussion.

Secondly, I recommend revision of adding one or more figures to the paper. I know the historic and paleontological significance of this work and it is a valued contribution, but I want a non-specialist to appreciate what's at stake here. I think one or two small figures would help. Something on the order of a cartoon that presents the premise at the beginning, and the conclusion at the end. Think of the figures that pepper papers from the Cretaceous work in the 1980s that depict the depth distributions of ammonites; these grab reader's attention and end up in text books. It's almost a "look what we can do! stick them in water profiles!" Ok, well, the author's discussion here handles many references to associations (or lack there of) between water/habitat depth, time, and suture complexity. Can we get a cartoon simplified illustration of this? Perhaps as a timeline, or as a water column profile? A little diagram like that might go a

long way in the mind of a reader, in terms of setting up the problem and providing your answer to it.

Overall the work is valuable and interesting, and I encourage the Editor to welcome minor revisions.

Author's Response to Decision Letter for (RSPB-2020-1919.R0)

See Appendix A.

RSPB-2020-1919.R1 (Revision)

Review form: Reviewer 1 (Christian Klug)

Recommendation

Accept as is

Scientific importance: Is the manuscript an original and important contribution to its field?

Excellent

General interest: Is the paper of sufficient general interest?

Good

Quality of the paper: Is the overall quality of the paper suitable?

Good

Is the length of the paper justified?

Yes

Should the paper be seen by a specialist statistical reviewer?

No

Do you have any concerns about statistical analyses in this paper? If so, please specify them explicitly in your report.

No

It is a condition of publication that authors make their supporting data, code and materials available - either as supplementary material or hosted in an external repository. Please rate, if applicable, the supporting data on the following criteria.

Is it accessible?

Yes

Is it clear?

Yes

Is it adequate?

Yes

Do you have any ethical concerns with this paper?

No

Comments to the Author

Hi,

I am happy with the ms.

Best wishes,

Christian Klug

Decision letter (RSPB-2020-1919.R1)

22-Sep-2020

Dear Dr Lemanis

I am pleased to inform you that your manuscript entitled "The ammonite septum is not an adaptation to deep water: re-evaluating a centuries old idea" has been accepted for publication in Proceedings B.

Open Access

Your article has been estimated as being 9 pages long. Our Production Office will be able to confirm the exact length at proof stage.

Paper charges

Sincerely,

Dr Daniel Costa
Editor, Proceedings B
mailto: proceedingsb@royalsociety.org

Appendix A

I thank both reviewers for taking the time to review this manuscript. I have implemented the small, grammatical changes and phrase changes through the text, mostly suggested by the first reviewer.

Below I will respond to the broader points brought up by both reviewers:

Reviewer 1:

“2. Page 4: You list 3 core components: this list lacks your suggestion from 2016 that the frilling may increase resistance against point loads by predators' teeth and jaws. It also lacks the null-hypothesis that the frilling is simply fabrication noise.”

I've made it clear in this part that the first component includes both hydrostatic pressure and point loads. The question that I'm specifically asking in this paper in relation to septa isn't: "What is the evolutionary significance of frilled septa". Instead it is "Can frilled septa strengthen the shell against water pressure". In this case I think the null-hypothesis would simply be "no".

“3. Why do you use the 9.11 mm for the cylinder?”

There is no particular reason for this. It turned out to be a convenient size considering the length of the digitized suture lines and after allowing for a small reduction in final circumference to create a bit of overlap where the two ends of the suture line could be joined.

“6. Last phrase of conclusion: While I agree with your conclusion, it would be helpful to know your opinion why ammonite sutures ARE frilled.”

This is THE question isn't it? Unfortunately I don't really have a strong answer for this, which is why I didn't include a section about this in the paper since it would just be idle speculation on my part. I think the potential for resisting point loads is still a compelling one that I am currently researching. Beyond this maybe some of the ideas of fluid flow around the chambers might have some merit but I think there needs to be more convincing research in that direction.

-“Potentially, it enables higher growth rates because the new septa that are not yet fully mineralized might be more tightly stayed when they are more strongly frilled than simpler septa. This is something that needs to be tested, maybe your next paper?”

Is this envisaging a system where the animal detaches from the septa prior to the completion of mineralization?

Reviewer 2:

“But... did ammonites rework, re-dissolve, re-mineralize, etc the aragonite within their shell interior surfaces?”

I added a small section to the discussion bringing this point up since I didn't really think of it while writing the original section but it's true that it is an underlying point in part of the argument.

I can make a long story short and say, no. There is no evidence for significant remodeling or resorption of the internal shell or shell features in ammonites specifically, or cephalopods in general. Working in a biomineralization group, I can say that there is a general appreciation for the fact that remodeling is more the exception than the rule and is not that common in a very general sense in molluscs. Functional remodeling, I think, is mostly limited to gastropods, especially of the internal shell features [1]. In cephalopods, the animal, specifically the mantle, can't access the earlier parts of the whorl due to the septa, which prevents later remodeling but also prevents the animal from repairing damage to the phragmocone. This is why the strength of the phragmocone is such an important topic when discussing shell mechanics in cephalopods. That isn't to say that resorption isn't seen at all, since new whorls overlap previous ones, there is some evidence in some groups of limited resorption of surface features—ornamentation—in this whorl overlap region [2,3].

1. Vermeij GJ. 2020 Overcoming the constraints of spiral growth: the case of shell remodelling. *Palaeontology*, 1–13. (doi:10.1111/pala.12503)
2. Signor PW. 1985 Surficial shell resorption in *Nautilus macromphalus* Sowerby, 1849. *The Veliger* **28**, 195–199.
3. Radtke G, Hoffmann R, Keupp H. 2016 Form and formation of flares and parabolae based on new observations of the internal shell structure in lycoceratid and perisphinctid ammonoids. *Acta Palaeontol. Pol.* **61**, 503–517. (doi:10.4202/app.00154.2015)

“Something on the order of a cartoon that presents the premise at the beginning, and the conclusion at the end. Think of the figures that pepper papers from the Cretaceous work in the 1980s that depict the depth distributions of ammonites; these grab reader's attention and end up in text books.”

This isn't a bad idea but I worry that the inclusion of this kind of cartoon might give people the wrong idea of what our conclusions actually are. We can't say anything directly about habitat depth, only about the mechanical consequences of the septa. It may be possible that some clades did live in deeper waters, but we can say that if this is true than that habitat transition was not facilitated by evolving complex septa. Of course, if the only argument for a deeper habitat is that the group has complex septa, than we can argue this is not a valid argument but that's really it.

We also can't limit habitat depth directly with the data presented here. But in the future, combining ongoing mechanical work and tomographic data, we could simulate the depth at which specific shells might break. In which case, then this kind of figure would be quite illustrative combined with that data.

Interestingly (perhaps), if you look at some of the older figures like this produced by Westermann, he does have a tendency to place most ammonoids in the upper 250 meters of the water column [1]

1. Westermann GEG. 1996 Ammonoid Life and Habitat. In *Ammonoid Paleobiology* (eds NH Landman, K Tanabe, RA Davis), pp. 607–707. Springer US.